SciPost Physics Proceedings

Submission

# Preliminary atmospheric effects through air showers at Agra using DEASA

Sonali Bhatnagar[1★], Shivam Kulshrestha[2]

**1** Faculty, Department of Physics and Computer Science Dayalbagh Educational Institute Dayalbagh, Agra, India
**2** Research Scholar,Department of Physics and Computer Science Dayalbagh Educational Institute Dayalbagh, Agra, India

★ sonalibhatnagar@dei.ac.in

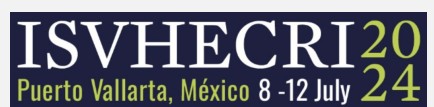

*22nd International Symposium on Very High Energy Cosmic Ray Interactions (ISVHECRI 2024) Puerto Vallarta, Mexico, 8-12 July 2024*

## Abstract

Investigations of the physical behaviour of the cosmic ray variations in various time scales are an important aspect in cosmic ray astronomy. The modulation of cosmic rays is an important tool for investigating disturbed behaviour in the heliosphere with longer time scales related to solar activities, while shorter time variations can be associated with Earth's atmospheric phenomena. The atmospheric temperature and pressure effect on count rates of DEASA detectors for 7 hours daily spanning 170 days from January to June 2022 is reported.The detectors are calibrated and their efficiencies have been plotted.Temperature and pressure profile at DEASA are studied.Then the cosmic ray intensities at one detector is studied to calculate the barometric and temperature coefficients.Finally the relative CR intensities of D6 detector is plotted with relative temperature and pressure in a time series plot.The graphs verify the expected behaviour of detector flux with atmospheric parameters and comparative study with other array data is reported.

# 1   Introduction

The dynamical state of the space weather and the atmospheric properties are encoded in the fluctuations of the cosmic ray flux measured. DEASA stands for Dayalbagh Educational Air Shower Array,is an educational experiment with the aim to study cosmic rays. The experiment is taking data with eight plastic scintillation detectors spread over an area of 260 sq.m. at Agra,near the Tropic of Cancer. These detectors have been operating since 2020 with an aim to promote astroparticle physics. The aim of this work is based on the importance of studying long time series of data giving correlation among cosmic rays, climate science and space weather. In this work, we show the beautiful symmetry of the behavior between the detector counts and atmospheric parameters.

In astroparticle physics, the study of the sun and its magnetic field added to the terrestrial magnetic field gives rise to fluctuations in the searches of the sources of cosmic rays [1]. The effect of solar phenomena on primary cosmic rays is limited to energies below 10 GeV. The solar wind has a strong stream of particles and its magnetic field affects the flux of primary cosmic rays. These solar wind particles are mostly electrons and protons with low energy (MeV). These solar particles interact with the terrestrial magnetic field in Van Allen belts which are extended over 2000 km to 15000 km for protons with intensities $10^8/cm^2sec$ and energies up to 1 GeV. The electron belt has intensities $10^9/cm^2sec$ at 3000 km and outer ring of 15000 km to 25000 km. Thus, these solar and Heliocentric processes affect the primary cosmic rays entering our atmosphere, which subsequently affects the climate and weather globally. In order to study cosmic rays' variations,the atmospheric effects on the ground array data has to be removed. This is the second order study to measure muon secondaries at the ground-based array. The initial studies are the variations in detecting counts vs pressure and temperature. The calibration of eight detectors is shown with their operating voltages. In section 3, the relative plots of detector count vs temperature and pressure are graphically represented. Finally, the coefficient of pressure and temperature are analysed for half year data.

In section 2, the calibration method for obtaining operating voltage for the PMT have been discussed and efficiency of detector is calculated. In section 3, weather studies have been performed and the graphs of atmospheric pressure, and temperature have been discussed. In section 4, the linear regression analysis has been done for relative pressure and temperature . The dependence of relative atmospheric pressure and atmospheric temperature on the relative intensity have been analysed.

# 2   DEASA: detector Studies

The DEASA detectors are calibrated every six months to observe the performance of all eight detectors,which are 100 cm × 100 cm × 2 cm plastic scintillator viewed by photomultiplier tube 9807B, located at a height of 65 cm above each detector inside the rain and weather

64 cover as shown in Figure 1.During the data period stated, their operating voltages were in
65 the range 1570 to 1630 V.The calibration plots of the observed counts of each detector vs
66 the corresponding voltage applied to that photomultiplier tube. The pulse height distribution
67 of each detector is observed for different voltages of PMT. The counts increase with voltage
68 and then become constant around the plateau which corresponds to operating voltage of the
69 detector for optimum performance.In this method [2] of calibration, plateau characteristic is
70 measured by keeping the paddle below the detector to observe the minimum ionizing particles
71 passing through both the detectors. The coincidence counts for a small-time window of 100
72 ns are observed by keeping a fixed gain voltage for the prototype detector placed below the
73 DEASA detectors. The applied voltage is varied from 1200 V to 1750 V to observe the plateau
region where the coincidence counts become independent and stable. Plastic scintillators have

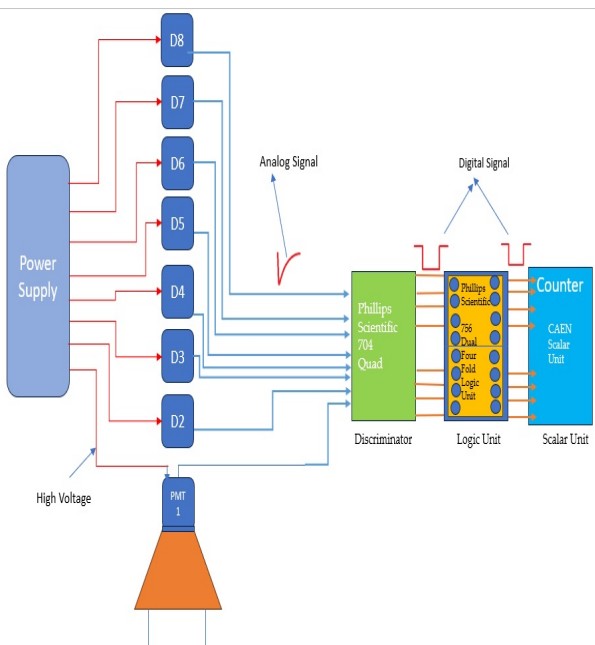

Figure 1: DEASA array

74
75 a small dead time, unlimited working life and high-count rate. Both the detector and PMT
76 are enclosed in a light aluminum cover along with a weather cover. The output of the eight
77 detectors is then sent to the nuclear electronics laboratory where the PMT signal is amplified
78 and fed to a discriminator which selects the signal from background noise. The signals are
79 fed to a logic unit and counter unit in the Nuclear Instrumentation Module (NIM) to collect
80 detector counts for predefined time. The hourly atmospheric pressure and temperature are
81 taken from the local weather data website. The pulse shape recorded on an oscilloscope have
82 amplitudes varying from 30mV to 80 mV for all the eight detectors. Preliminary results show
83 that all eight detectors, arranged at the same horizontal level exhibit varying performance.
84 The efficiency of each detector has been computed and shown for four of them in figure 2.
85 The range of the efficiency of the detector varies for each detector D1, D2, D3, D4, D5, D6,
86 D7, and D8 as 0.85, 0.78, 0.65, 0.76, 0.84, 0.89, 0.95, and 0.75 respectively during the period
87 stated.

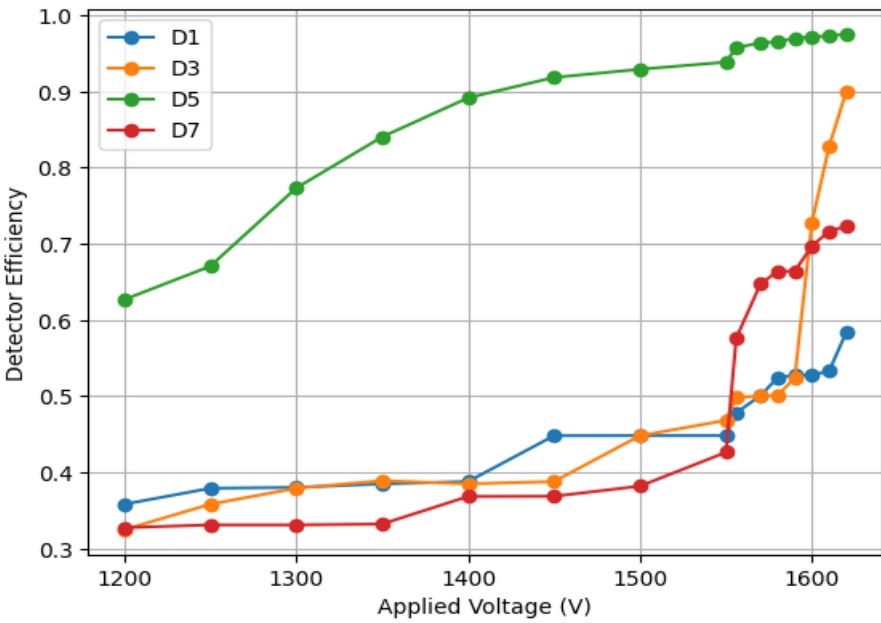

Figure 2: Efficiency graph for four detectors in DEASA array

## 3 Variation in cosmic ray intensities with atmospheric parameters

The secondary cosmic rays measured at the ground-based arrays show temporal variations due to atmospheric fluctuations [3]. For ground detectors, the main atmospheric effects are the pressure ,relative humidity and temperature.The count rate for the eight detectors was observed hourly from morning 10 am to evening 5 pm every day from January 2022 to June 2022. In the table 1 below, the minimum, maximum, mean, skewness and kurtosis for pressure, temperature, and detector count rates are shown.

|  | Min. | Max. | Mean | Skewness | Kurtosis |
|---|---|---|---|---|---|
| Pressure (in mb) | 970 | 1020 | 1002 | - 0.91 | - 1.8 |
| Temperature (in C) | 11 | 41 | 24 | -0.4 | -1.05 |
| Counts/min | 4504 | 16290 | 10397 | -1.22 | 1.23 |

Table 1: **Summary of the mean values of the parameters**.

The skewness is measure of how much a random probability distribution varies from the normal distribution. The negative value of skewness indicates that the left tail of the distribution is relatively longer.The kurtosis is the measure of "tailedness" of the probability distribution of a real-valued random variable. The negative value indicates a distribution which is more peaked than normal and positive value of kurtosis indicates a shape flatter than normal[4]. The figure 3 depict the pressure and temperature measured at detector 6 (D6) in the array. The graph indicate that barometric pressure (in mbars) which is decreasing with time scale, atmospheric temperature increases on the same scale from January 2022 to June 2022.

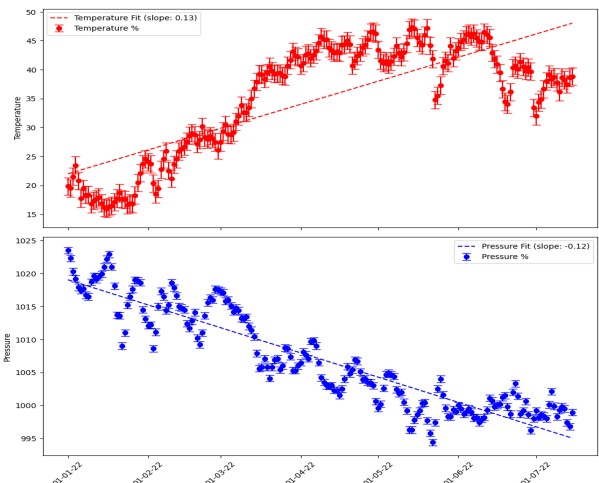

Figure 3: Pressure-Temperature profile at DEASA

## 4   Analysis

The pressure effect on secondary cosmic rays has been studied by the equation:

$$\frac{\Delta I}{I} = \beta \Delta P \tag{1}$$

$\frac{\Delta I}{I}$ is the deviation of the cosmic ray flux and $\Delta P$ is the deviation in atmospheric pressure and $\beta$

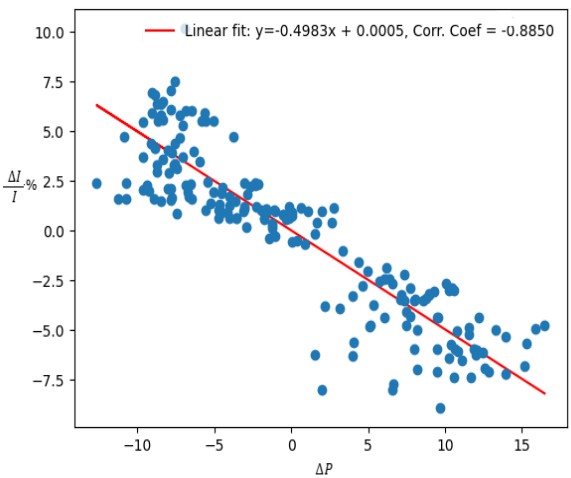

Figure 4: CRs intensity vrs Pressure at D6

is the barometric coefficient which depends on other factors like the identified secondaries and the altitude above sea level where the experiment is performed. The atmospheric temperature also affects the flux of the cosmic ray detected at the ground arrays; this seasonal variation observes a maximum and minimum in summer, winter corresponding to positive and negative coefficients. The positive effect is related to the influence of temperature on the residual muon from the charge pion decays[5] and the temperature coefficient is represented by:

$$\frac{\Delta I}{I} = \alpha \Delta T \tag{2}$$

where, $\frac{\Delta I}{I}$ is the deviation of cosmic ray flux and $\Delta T$ is deviation in atmospheric temperature and $\alpha$ is the temperature coefficient. It is observed from fig. 4, the relative intensity

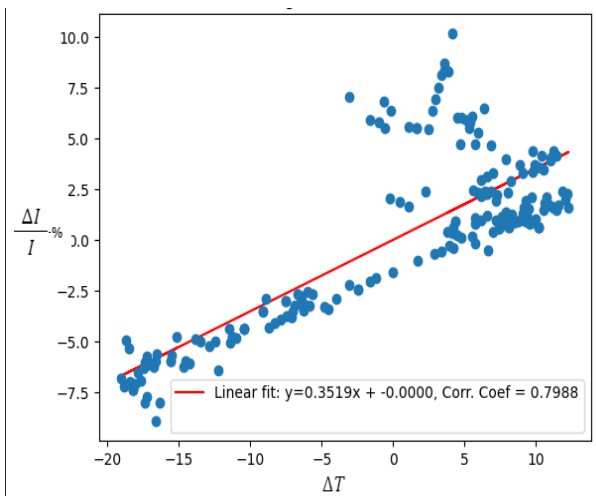

Figure 5: CRs intensity vrs temperature at D6

of secondary count rate falls from January 2022 to June 2022. The relative pressure shows
a negative effect from January 2022 to June 2022.The figure 5 depicts variation in relative
intensity of secondary count rate with relative temperature with a positive effect from January
2022 to June 2022.The values of slope obtained from the graph between the relative count
rate vs relative pressure is -0.49 /mbar with a correlation coefficient of -0.89.The slope for the
relative count rates vs relative temperature is $0.35/°C$ and correlation coefficient 0.79. The
plot of Figure 6 between relative intensity, relative pressure and temperature from January to
June 2022 are plotted for detector D6.The graphs between relative detector count rate of sec-
ondary cosmic rays per minute show correlation with temperature and anti-correlation with
pressure.The observed barometric coefficient is negative indicating that count rates are in-
creasing with decrease in pressure and temperature coefficient is positive indicates that count
rates are increasing proportionally with temperature. Further the analysis has been done to

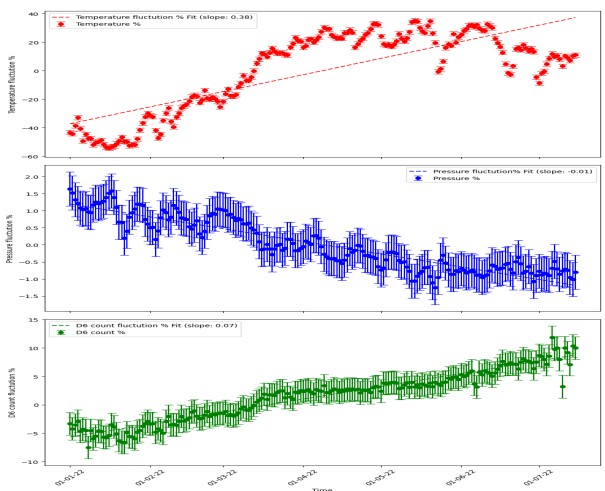

Figure 6: Pressure-Temperature profile at D6 detector

study the variations between the above two atmospheric parameters and the relative detector
count rate.

## 5   Conclusion

The results of the cosmic rays and atmospheric temperature data with a cosmic ray detector CARPET at San Juan, Argentina, 31 S, 69 W,2550 m over sea level with geomagnetic rigidity cutoff RC 9.8 GV. They found an anti correlation between the relative variations in intensity of the cosmic rays and surface temperature at an altitude of maximum production of air shower secondaries. The variation between cosmic rays' intensity and atmospheric pressure shows anti correlation, and the barometric coefficient is found to be -0.44/mbar and the tempera-ture coefficient is -0.4/°C [6]. The results obtained by a group from KACST detectors (Riyadh, Saudi Arabia; RC = 14.4 GV) of count rates vs temperature shows positive correlation with temperature coefficient is +0.04 /°C and pressure coefficient found to be -0.15/mbar for de-tector of dimension 1 $m^3$ [7]. It is observed from our results in comparison to other results from literature, the cosmic ray flux variation with temperature shows positive correlation as well as negative correlation depending on the geographical location, altitude, etc. The detailed investigation will be presented in the coming future.

## Acknowledgment

The authors are thankful to the Dayalbagh Educational Institute for funding DEASA. The au-thors are thankful to the Cosmic ray laboratory, Ooty, T.I.F.R. in the development of DEASA.

**Funding information**   The authors acknowledge the financial grant by Dayalbagh Educa-tional Institute,Agra for funding DEASA.

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
