# Peer review of "Preliminary atmospheric effects through air showers at Agra using DEASA"

_SciPost Physics Proceedings_

## Round 1 · Referee Report · Anonymous (Referee 1) · 2025-2-2

Strengths

The content of the article

Weaknesses

The english writing. It is acceptable, but can be better.

Report

This article gives a good overview of the DEASA (array) experiment in Agra, focusing on atmospheric effects and air showers. The work is of an academic standard and deserves publication. However, some actions are needed before publication. These recommendations aim to improve the manuscript according to the rules and quality of the journal.

Requested changes

1) The abstract should fit in 8 lines. It should be written in a clear and understandable style, emphasizing the context, the problem(s) studied, the methods used, the results obtained, the conclusions reached, and the outlook.

2) English must be improved. The reviewer strongly recommends having a native English speaker read through the paper to improve the English. Use the active voice in every sentence that focuses on the doer. Many passive sentences cloud the meaning of the sentences. The reviewer recommends using the active voice whenever possible. Sentences in the passive voice often use more words, can be vague, cloud the meaning and lead to a jumble of prepositions. The active voice emphasizes the subject, i.e. the person performing the action. In the passive voice, on the other hand, the action or the recipient of the action is emphasized

3) The authors must cite the figures as the source and give the name in all illustrations. For example: Credit AUTHOR (REFERENCE). Due to editorial rules and copyrights, citing sources (see References) and giving credit is necessary. It is best to use typical rules such as "modified by xxx(yyyy)" or "based on xxx (yyyy)", "Adapted from [reference to the original article], "with permission from [copyright holder]", or similar, depending on the context. Also, use Creative Commons Attribution 4.0 International (CC-BY-4.0) if appropriate. ( https://creativecommons.org/licenses/by/4.0/ ).

Besides, the labels in Figures 3 and 6 are tiny; authors must make them bigger.

4) The captions must be brief but comprehensive, but no shorter than in the manuscript. The reviewer suggests making them more descriptive. The caption should describe the data shown, draw attention to important features within the figure, and may sometimes include interpretations of the data.

5) Check and insert spaces between punctuation marks, such as the period and the next word. For example, check line121

6) Insert a comma at the end of equations (1) and (2). Equations must be punctuated.

7) In line 97, the sentence:

The kurtosis is the measure of "tailedness" of the probability distribution of a real-valued random variable.

must be rephrased because plagiarism detection is based on the next source:

Almaiah, M. A., Jalil, M. @. M. A., & Man, M. (2016). Empirical investigation to explore factors that achieve high quality of mobile learning system based on students' perspectives. https://doi.org/10.1016/j.jestch.2016.03.004

8) In the conclusion section:

-- The sentence "The results of the cosmic rays and atmospheric temperature data with a cosmic ray detector CARPET at San Juan, Argentina, 31 S, 69 W, 2550 m over sea level with geomagnetic rigidity cutoff RC 9.8 GV." must be rewritten as it appears to be a sentence fragment and is not clear.

-- The sentence: "The detailed investigation will be presented in the coming future" needs to be rewritten because the plagiarism detection of:

Matsuda, M., & Iino, M. (1969). The Solvent Effect on the Composition of Styrene Polysulfone. Macromolecules. https://doi.org/10.1021/ma60008a024

9) Check for coherence when referring to figures and tables. The reviewer suggests using a Table, Figure, Section, and Equation words. For example, compare lines 113, 115 and 120; fig. 4, figure 5, Figure 6

Recommendation

Ask for minor revision

---

## Editorial Decision

in_refereeing